# The Type IX Secretion System: Advances in Structure, Function and Organisation

**DOI:** 10.3390/microorganisms8081173

**Published:** 2020-08-01

**Authors:** Dhana G. Gorasia, Paul D. Veith, Eric C. Reynolds

**Affiliations:** Oral Health Cooperative Research Centre, Melbourne Dental School, Bio21 Institute, The University of Melbourne, Melbourne VIC 3010, Australia; gorasiad@unimelb.edu.au (D.G.G.); pdv@unimelb.edu.au (P.D.V.)

**Keywords:** bacterial protein secretion, type IX secretion system, cell-surface attachment, *Porphyromonas gingivalis*, Flavobacterium johnsoniae

## Abstract

The type IX secretion system (T9SS) is specific to the *Bacteroidetes* phylum. *Porphyromonas gingivalis*, a keystone pathogen for periodontitis, utilises the T9SS to transport many proteins—including its gingipain virulence factors—across the outer membrane and attach them to the cell surface. Additionally, the T9SS is also required for gliding motility in motile organisms, such as *Flavobacterium johnsoniae.* At least nineteen proteins have been identified as components of the T9SS, including the three transcription regulators, PorX, PorY and SigP. Although the components are known, the overall organisation and the molecular mechanism of how the T9SS operates is largely unknown. This review focusses on the recent advances made in the structure, function, and organisation of the T9SS machinery to provide further insight into this highly novel secretion system.

## 1. Introduction

The *Bacteroidetes* are an important group of bacteria found in the digestive tract of animals and humans and are also widespread in the environment. Many members of this phylum utilise the type IX secretion system (T9SS) to secrete various effector proteins, such as proteases, adhesins, cellulases, chitin and surface layer proteins [1,2,3,4,5,6]. The T9SS is important in the periodontal pathogen, *Porphyromonas gingivalis*, for the secretion of at least 30 proteins, including its major virulence factors called gingipains (Kgp, RgpA, RgpB) [7,8,9]. It is also responsible for major bacterial diseases in birds and farmed fish, such as rainbow trout fry syndrome and cold-water disease [10,11,12,13]. Interestingly, the T9SS is also required for gliding motility and *Flavobacterium johnsoniae* is used as a model organism to study this kind of motility [7,14]. SprB and RemA, the major motility adhesins that are required for gliding motility in *F. johnsoniae*, are delivered to the cell surface by the T9SS [7,15].

Protein substrates (cargo proteins) of the T9SS have N-terminal signal peptides for transport across the inner membrane by the Sec system and are targeted to the outer membrane translocon via their conserved C- terminal domain signal (CTD) of approximately 80 amino acid residues [16,17,18,19]. The CTD signal has been identified to be of two types, type A and type B [19]. In *P. gingivalis*, all cargo proteins except PG1035 have a type A CTD. The T9SS is composed of at least 19 protein components, namely: PorE, PorF, PorG, PorK/GldK, PorL/GldL, PorM/GldM, PorN/GldN, PorP, PorQ, PorT/SprT, PorU, PorV, PorW/SprE, PorZ, Sov/SprA, and Plug as well as the three transcription regulators PorX, PorY and SigP [7,20,21,22,23,24,25,26,27,28,29,30]. More recently, PorA and PGN_1783 have also been identified as new components of this system, which are discussed below.

In *P. gingivalis*, the secreted type A cargo proteins bind to the outer membrane protein PorV and are transported or shuttled to the attachment complex comprising of PorU, PorZ, PorQ and additional PorV [31]. The PorU sortase cleaves the CTD signal on the cell surface and conjugates the cargo to the A-LPS anchor to produce the virulence coat, also known as the electron-dense surface layer [20,21,32]. Until recently, the functions of all other components of the T9SS were largely unknown, however, their locations within the cell envelope were known (Table 1). PorL and PorM are embedded in the inner membrane [7]; PorF, PorG, PorP, Sov, PorV, PorQ and PorT form outer membrane beta-barrel structures [20,23,26,33]; PorK, PorE and PorW are lipoproteins associated with the outer membrane and localised in the periplasm [7,22,23]; PorU and PorZ are cell surface proteins anchored by PorV and PorQ [31], respectively; PorN is a periplasmic protein anchored to the outer membrane via PorK [23]. Two recent comprehensive reviews on the T9SS describe all the protein components that have been identified to date [1,2]. This current review will focus on the most recent advances made in the elucidation of the structure, function and organisation of the known components of the T9SS machinery.

## 2. Structure

### 2.1. SprA/Sov: The Translocon

The translocon of the T9SS was recently identified in *F. johnsoniae* as SprA (*P. gingivalis* Sov homolog), based on the structural evidence [28]. SprA was isolated from *F. johnsoniae* using a twin-Strep-tag purification system and it was found as two separate complexes, one bound to PorV and peptidyl-prolyl cis-trans isomerase (PPI) and the other with a novel component, named the Plug and PPI (Figure 1) [28]. High-resolution cryo-electron microscopy (cryo-EM) of these two complexes revealed that SprA forms a 36-stranded outer membrane beta-barrel structure [28]. This is the largest outer membrane beta-barrel structure ever identified. The internal diameter of the SprA translocon was found to be ~7.0 nm, large enough to accommodate folded cargo proteins. The barrel is capped at the cell surface, open at the periplasmic end, and has a lateral opening in the barrel wall. PorV blocks the lateral opening and, interestingly, one of its surface loops is located inside the SprA translocon. In this complex, the periplasmic pore is open (Figure 1A), while in the SprA-Plug complex the periplasmic opening is occluded by the Plug, and SprA is open at the lateral side (Figure 1B) [28]. The binding of PorV and the plug to SprA is mutually exclusive. These structures demonstrate a pathway for substrates to pass through the outer membrane. In the PorV-bound state, the channel is open to allow the uptake of T9SS substrates from the periplasm, while the plug complex represents the channel after substrate has been released. Since PorV has been shown to associate with the T9SS substrates on the cell surface [31], the authors hypothesize that PorV is able to bind to the substrates in the interior of the SprA barrel and trigger the release of PorV from the SprA channel [28]. The Plug protein then binds to the SprA at the periplasmic end to seal the channel. Together, this provides unidirectional transport of the T9SS substrates [28]. Although the structure of Sov is not solved, it is confidently modelled to have a very similar structure to SprA (Figure 1C). We have recently shown that Sov associates with either PorV or the plug in a similar manner to SprA [34] (non-peer-reviewed)and this interaction is discussed below in greater detail. 

### 2.2. PorL and PorM: Molecular Motors

PorL (GldL) and PorM (GldM) are the only members of the T9SS that are localised in the inner membrane and have been shown to form a complex [7,23,36]. PorL is known to have two transmembrane helices and a cytoplasmic domain, while PorM has one transmembrane helix and a periplasmic region [36]. They are known to interact via their transmembrane segments [36]. The atomic structures of the periplasmic regions of both PorM and GldM were solved and shown to exist as dimers and consist of four domains, D1 to D4 (Figure 2). GldM crystallised readily, however, full- length PorM resisted crystallisation and therefore fragments of PorM were crystallised and assembled based on GldM structure. The total length of the D1 to D4 domains was approximately 180-Å and therefore can span most of the periplasmic space [37,38]. Domains D1 to D4 of GldM shared the same straight topology, whereas in PorM there was ~ 45° bend between domain D2 and D3 [37]. Based on the observation of SprB circular motion in sheared cells [39] and the localisation of YFP-labelled GldL close to the centre of that rotation, Shrivastava et al. [40] recently predicted GldLM to be the motor of the T9SS and showed that it was driven by the proton motif force (PMF). This was confirmed by Hennell-James et al. [41] (non-peer-reviewed), by solving the native structure of the GldLM /PorLM complexes using cryo-EM. The proteins were expressed and purified from *Escherichia coli*. The complex comprising GldL and truncated GldM was found to be asymmetric with GldL, forming a distorted ring composed of five subunits in the inner membrane, and the GldM D1 domain emanating from the ring with its transmembrane helices completely buried in the proteinaceous environment of the ring, with no exposure to the lipid environment (Figure 2) [41]. In the full length PorLM native structure, the bend within the PorM shaft was observed to be between domains D1 and D2 [41], as opposed to domains D2 to D3 in the crystal structure. This discrepancy is likely due to an artefact in the crystal structure produced by expressing PorM in truncated forms. The periplasmic domains of the GldM subunits were packed against the top of the ring of GldL transmembrane helices leaving no room for proton movement, which implies that conformational changes are required to open an aqueous channel in the ring to allow proton movement. Using substitution experiments, it was shown that protonation of aromatic residues GldLY13 and GldMY17 was important for their function and perhaps involved in transducing transmembrane proton movements to mechanical work [41]. Together, the findings suggest that GldL/PorL is a stator, and proton flow causes the GldM/PorM dimer (rotor) to rotate in a wide circle of approximately 18 nm diameter, which would in turn drive processes at the outer membrane, such as gliding and protein transport. It has been shown that domain D4 of PorM interacts with PorN, whereas domains D2, D3 and D4 interact with the PorK lipoprotein [37]. This suggests that the energy may be transduced to the outer membrane via GldK/PorK and GldN/PorN. It remains to be elucidated how many GldLM rotary motors are located along the motility track in *F. johnsoniae*, and how many are present per PorK/N ring in *P. gingivalis*. Furthermore, there is little understanding at present as to how this rotary motor drives T9SS cargo protein secretion and forms the surface layer in *P. gingivalis.*

### 2.3. PorE

In 2016, PorE was shown to be a new component of the T9SS [22]. Its sequence analysis revealed that it is a lipoprotein and is composed of four domains: a tetratricopeptide repeat domain; a five-bladed beta-propeller domain; a predicted carboxypeptidase regulatory domain-like fold, and an OmpA_C-like domain [22]. A crystal structure of the OmpA_C-like domain bound to the peptidoglycan was solved recently [42]. This domain (residues 534 to 668) was expressed and purified from *E. coli*. The solved structure was a tetramer; however, the oligomerization was perhaps an artefact of crystallization as size exclusion chromatography, and multi-angle light scattering analysis indicated that the protein was stable in a monomeric form in solution [42]. The structure consists of a three-stranded beta-sheet and five alpha-helices with the connectivity of α1β1α2β2α3α4α5β3. Three residues (Asp576, Asn584 and Arg591) form hydrogen bonds with the peptidoglycan; these residues are known to be conserved in the OmpA family. To date, PorE is the only T9SS component that is known to associate with the peptidoglycan, and it has been suggested that the role of PorE may be to anchor the T9SS to the peptidoglycan and act as a template for the assembly of the translocon. Further work is needed to confirm this suggestion, and it would also be interesting to see if there are other components of the T9SS that associate with the peptidoglycan.

### 2.4. In-Situ Structure of the PorK/N Complex

Our previous studies showed that the PorK lipoprotein and PorN form a very large ring complex, measuring 50 nm in diameter, which localised to the periplasmic face of the outer membrane [23]. The existence of this large ring structure in the cell was questioned by Leone et al. [37], where the authors suspected that the rings were an artefact of the purification method. Using electron cryotomography (ECT) of whole *P. gingivalis* cells, we recently showed that the rings are indeed present in cells [34]. The ring structures observed in-vivo were very similar in size and shape to the purified PorK/N rings [34]. Furthermore, these structures were absent in the *porK* mutant, confirming that this protein was also involved in the formation of the rings observed in-vivo [34]. In our in-vitro analysis, we had predicted that the rings were tethered to the outer membrane, and located in the periplasm [23]. This was confirmed in the ECT images, in which the rings were found immediately underneath the outer membrane. Unfortunately, no other clear structures were observed in the ECT images, suggesting structural or compositional heterogeneity. Although there are homologs of PorK and PorN present in *F. johnsoniae*, termed GldK and GldN, it remains to be determined whether GldK and GldN form the same large ring-like structures in this organism.

## 3. Architecture

### Organisation of the T9SS

Although the components of the T9SS have been known for over a decade, there is a large gap in our understanding of how they are architecturally organised to form the T9SS. A dense network of interactions between PorK, PorL, PorM, PorN and PorP has been shown by Vincent et al. [36]. However, it was unclear how these components interacted with the rest of the T9SS. Our recent comprehensive proteome interactome analysis of the T9SS components provided significant insights into the organisation of this novel system [34]. The Sov translocon was found to associate with the PorK/N rings via the PorW protein and another new component PGN_1783 (Figure 3). Importantly, this arrangement links the translocon all the way through to the PorL/M rotary motor. Additionally, Sov was also found to be in complex with either PorV and a novel component PorA or with the Plug protein [34]. Since the interactions reported between SprA-PorV and SprA-plug in *F. johnsoniae* were also found with the *P. gingivalis* orthologue Sov, it is very likely that the mechanism of substrate translocation is the same in both species. A separate complex consisting of PorP, PorE and PG1035 was also identified [34]. PG1035 is the only type B CTD protein present in *P. gingivalis,* while in motile organisms such as *F. johnsoniae*, there are multiple proteins with a type B CTD. SprB, the major protein involved in gliding motility in *F. johnsoniae* is a type B CTD protein, and we showed it to anchor on to the cell surface via SprF, a PorP-like protein [34]. Since SprB translocation is independent of PorV [43], this led to the hypothesis that PorP-like proteins are able to collect their respective substrates directly from the translocon in a similar manner to PorV [34]. We propose that the PorP-PorE-PG1035 complex functions in anchoring the T9SS to the peptidoglycan wall. It is known that the attachment complex is composed of PorV, PorU, PorZ and PorQ and this complex was not found to associate with other components [31]. Since the in-situ structure of the PorK/N ring was found to be tightly bound to the outer membrane, we have proposed that a function of the PorK/N rings is to form a barrier to localise all the outer membrane T9SS components in an outer membrane island to harmonise secretion and cell surface attachment of the cargo proteins [34]. Further analysis using ECT and super-resolution microscopy may shed more light on the precise organisation and architecture of the T9SS. 

## 4. T9SS Substrate Route

### Substrate Translocation Pathway

Although there has been significant work on identifying the components of the T9SS and understanding their roles, very little research has been performed on their interactions with the cargo proteins. Previously, it has been shown that PorV binds to the cargo proteins and is thought to shuttle them to the attachment complexes [31]. One recent study used a two-hybrid system to shed further light on cargo-component interactions [44] (non-peer-reviewed). It was found that the cargo proteins associated with PorN and three periplasmic domains of PorM, but not PorK [44]. Presumably, these protein components have different affinities to the CTD signal, therefore allowing the sequential passage from one component/domain to the next. These were the only three protein components that were tested in this study. Further work on analysing other components, such as PorW will greatly assist in the understanding of how the cargo proteins are recruited and transported across the outer membrane and finally anchored on to the cell surface via A-LPS. 

## 5. Conclusions and Future Directions

The novel T9SS described here is remarkably complex. It allows the secretion of very large proteins, which are important for pathogenesis, gliding motility and the digestion of cellulose and chitin. Recent structures of the translocon, molecular motors and other components have significantly advanced the knowledge in this field. The protein interactome study has provided major insights into the organisation of the T9SS, and shows how the motors may be connected to the translocon (Figure 3). Further studies on how the energy is transduced from PorM via the PorK/N rings to Sov may provide critical mechanistic information on how the T9SS functions. 

## Figures and Tables

**Figure 1 microorganisms-08-01173-f001:**
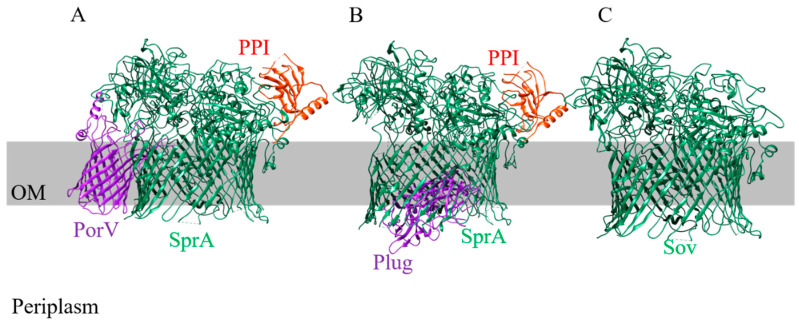
Cryo-EM structure of SprA complexes. (**A**) SprA bound to PorV at the lateral opening. (**B**) SprA bound to the Plug at the periplasmic opening. (**C**) Phyre 2 modelled structure of Sov [35]. The PDB entries (www.rcsb.org) for the structures shown are: SprA-PorV-PPI (6H3I) and SprA-Plug-PPI (6H3J). OM: outer membrane

**Figure 2 microorganisms-08-01173-f002:**
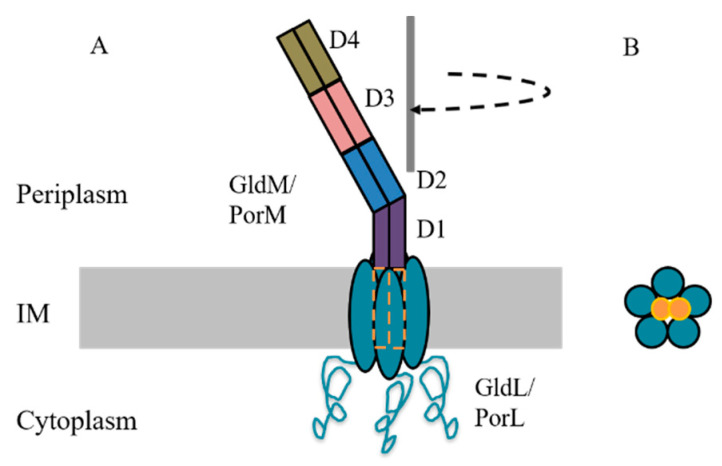
Cartoon representation of the GldL/M motor complex. (**A**) GldL/PorL forms a pentameric ring in the inner membrane. A dimer of GldM/PorM is inserted inside the GldL/PorL ring and domains D1 to D4 span the periplasm. The bend between D1 and D2 allows it to rotate in a wide circle. (**B**) Cross-section view showing pentameric arrangement of GldL/PorL and packed GldM/PorM helices in the middle. Figure adapted from [41]. IM: inner membrane

**Figure 3 microorganisms-08-01173-f003:**
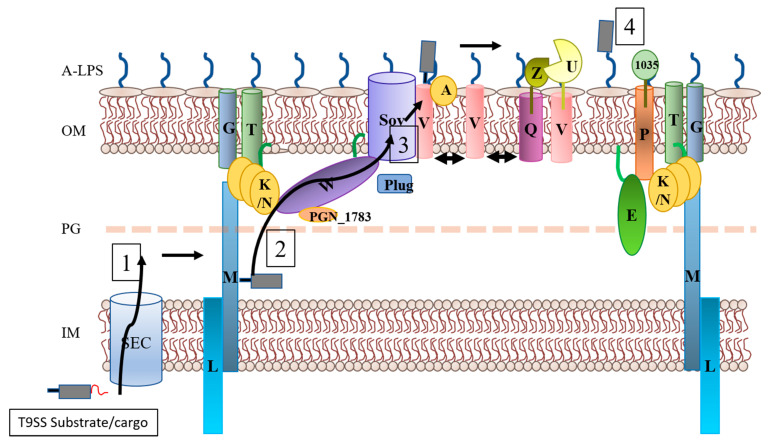
The proposed model of the T9SS. The T9SS substrates (cargo proteins) contain an N-terminal signal (shown as a red line) that allows their passage through the SEC complex. (1) The N-terminal signal is cleaved off and the T9SS substrates are recruited to the T9SS translocation system. (2) The substrates bind to PorM via their C-terminal domain (CTD) signal (black line) and, using this signal, they get passed on to PorN and PorW until they reach the Sov translocon. (3) The substrates are collected by PorV from the Sov channel and PorV shuttles them to the attachment complex consisting of PorU, PorV, PorQ and PorZ, whereby a sortase-like reaction occurs in which the CTD signal is swapped with A-LPS. (4) The substrates are anchored covalently to A-LPS on the cell surface and the CTD signal is released into the culture fluid.

**Table 1 microorganisms-08-01173-t001:** Localisation and proposed function of the T9SS protein components.

Pg	Proposed Function	Localisation	Fj
PorE	Binds to the peptidoglycan, PorP and PG1035	OM lipoprotein	many
PorF	Unknown	OM	-
PorG	Binds to the PorK/N rings	OM	-
PorK	Forms ring structure with PorN	OM lipoprotein	GldK
PorL	Motor stator	IM	GldL
PorM	Rotor and shaft/pinion	IM	GldM
PorN	Forms ring structure with PorK	OM/Periplasm	GldN
PorP	Binds to PorE and PG1035	OM	many
PorQ	Anchor for PorZ	OM	PorQ
PorT	Unknown	OM	SprT
PorU	Sortase	Cell surface	PorU
PorV	Shuttle protein for type A cargo proteins	OM	PorV
PorW	Unknown	OM lipoprotein	SprE
PorZ	Involved in the cell surface anchorage of type A cargo proteins	Cell surface	PorZ
Sov	Translocon	OM	SprA
-	Binds to SprB motility adhesin	OM	SprF

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
