# Peer review of "The Type IX Secretion System: Advances in Structure, Function and Organisation"

_microorganisms, 2020, doi:10.3390/microorganisms8081173_

Round 1

Reviewer 1 Report

In this review, Gorasia et al. describe the most recent findings concerning the T9SS, mostly at the structural level. This represents a valuable work as many structures were described in the last few years, and permits a better representation of the highly complex T9SS organization.

I have only one critical: several references (34, 40, 43) are related to results that were not submitted to peer-review and I consider that it should be mentioned. Indeed, peer review is a process that cannot be ignored to guarantee a scientific production of quality.

Therefore, I agree with the publication of the manuscript, with the mentioned minor revision.

Author Response

I have only one critical: several references (34, 40, 43) are related to results that were not submitted to peer-review and I consider that it should be mentioned. Indeed, peer review is a process that cannot be ignored to guarantee a scientific production of quality.

We have now mentioned “non-peer review” immediately after these references are cited in the revised manuscript (pg 5, 6, 11).

Reviewer 2 Report

Gorasia and co-workers summarize our recent understanding of the T9SS. The review is well written. However, some references are misplaced or missing. Minor comments are below.

  1. Literature suggests that besides gingivalis, T9SS proteins are generally named with the Gld and Spr nomenclature. . For eg. Gldk/PorK, GldL/PorL, GldM/PorM, GldN/PorN, SprT/PorT, SprT/PorT, SprA/Sov. While this is pointed out in Table 1, in order to make this review easy for read for the general reader, I suggest that the authors use both nomenclatures while introducing the proteins on page1.
  2. On multiple occasions, the authors point towards rotation of the T9SS. However, the article (Shrivastava, Lele, Roland, Berg, Current Biology, 2015) that demonstrated rotation is never cited.
  3. Reference#39 is misplaced. The correct reference for the sentence taken verbatim from the review ‘Based on the observation of SprB circular motion in sheared cells and the localization of YFP-labelled GldL close to the center of that rotation…..’ should be Shrivastava and Berg, Science Advances, 04 Mar 2020:Vol. 6, no. 10, eaay6616. It should be noted that this study demonstrated that rotation is driven by a proton motive force i.e. addition of CCCP stopped rotation and resurrection of the motor occurred after washing the CCCP off. Later, Hennell James et al. (ref#40) showed that Type 9 protein transport is protonmotive force-dependent i.e. addition of CCCP blocked transport of a tripartite fusion composed of a N-terminal SS, a mCherry passenger domain, and a C-terminal T9SS CTD
  4. ‘In the full length PorLM native structure, the bend within the PorM shaft was observed to be between domains D1 and D2…….’ Please provide reference.
  5. The pictures of PorK/GldK and PorN/GldN ring that Gorasia et al. presented in the past are fascinating. However, a size of this ring (about 50 nm diameter) appears to be large. Maybe the authors can predict the number of GldM monomers that would fit on a ring of this size? If yes, a predictive cartoon of how a GldK/N ring might fit in the current picture of T9SS/gliding machinery of F. johnsoniae would make this review informative and memorable.

Author Response

  1. Literature suggests that besides gingivalis, T9SS proteins are generally named with the Gld and Spr nomenclature. . For eg. Gldk/PorK, GldL/PorL, GldM/PorM, GldN/PorN, SprT/PorT, SprT/PorT, SprA/Sov. While this is pointed out in Table 1, in order to make this review easy for read for the general reader, I suggest that the authors use both nomenclatures while introducing the proteins on page1.

Both nomenclatures have now been used on page 2 of the revised manuscript.

  1. On multiple occasions, the authors point towards rotation of the T9SS. However, the article (Shrivastava, Lele, Roland, Berg, Current Biology, 2015) that demonstrated rotation is never cited.

This article that first demonstrated a rotary motor is now cited in the revised manuscript (pg 6).

  1. Reference#39 is misplaced. The correct reference for the sentence taken verbatim from the review ‘Based on the observation of SprB circular motion in sheared cells and the localization of YFP-labelled GldL close to the center of that rotation…..’ should be Shrivastava and Berg, Science Advances, 04 Mar 2020:Vol. 6, no. 10, eaay6616. It should be noted that this study demonstrated that rotation is driven by a proton motive force i.e. addition of CCCP stopped rotation and resurrection of the motor occurred after washing the CCCP off. Later, Hennell James et al. (ref#40) showed that Type 9 protein transport is protonmotive force dependent i.e. addition of CCCP blocked transport of a tripartite fusion composed of a N-terminal SS, a mCherry passenger domain, and a C-terminal T9SS CTD

        This reference [39] has been corrected and the sentence “…..rotation is    driven by the proton motif force” has been included in the revised manuscript (pg 6)

  1. ‘In the full length PorLM native structure, the bend within the PorM shaft was observed to be between domains D1 and D2…….’ Please provide reference.

A reference has been added to substantiate this statement in the revised manuscript (pg 6).

  1. The pictures of PorK/GldK and PorN/GldN ring that Gorasia et al. presented in the past are fascinating. However, a size of this ring (about 50 nm diameter) appears to be large. Maybe the authors can predict the number of GldM monomers that would fit on a ring of this size? If yes, a predictive cartoon of how a GldK/N ring might fit in the current picture of T9SS/gliding machinery of F. johnsoniae would make this review informative and memorable.

The large (50 nm) PorK/N ring structures have only been demonstrated in P. gingivalis. It is unknown if 50 nm GldK/N ring structures are also present in F. johnsoniae. In fact initial attempts have failed to identify them in this gliding bacterium. The rotary motor in F. johnsoniae has been linked with moving an adhesin around the length of the cell to allow gliding motility. P. gingivalis lacks this function and presumably uses the rotary motor and the rings to spin out a surface layer that is not present in F johnsoniae. Hence there are clear differences in function of the T9SS and accessory proteins in these two bacterial species. Therefore, including a cartoon of GldK/N rings with the gliding machinery of F johnsoniae would be very speculative and thus not suitable for a review on the T9SS at this stage.

Additional corrections:

Three additional changes have been made:

Table 1 pg 4: Coupling has been changed to Shuttle

Pg 6: Periplasmic domain has been changed to periplasmic region

Pg 6: Hennel-James also demonstrated that T9SS protein transport is energised by the proton motif force has been deleted.

Pg 7: GldK and GldN are deleted and included later in the sentence.